# Pediatric Mixed-Phenotype Acute Leukemia: What’s New?

**DOI:** 10.3390/cancers13184658

**Published:** 2021-09-16

**Authors:** Sandeep Batra, Anthony John Ross

**Affiliations:** Department of Pediatrics, Indiana University School of Medicine, Riley Hospital for Children at Indiana University Health, Indianapolis, IN 46202, USA; rossant@iu.edu

**Keywords:** mixed-phenotype acute leukemia, MPAL, *Ph*+, *MLL*, lineage switch, minimal residual disease, treatment

## Abstract

**Simple Summary:**

Pediatric mixed-phenotype leukemia is a rare form of blood cancer in children. In this review, we cover both the evolution of treatment over the past several years and outline new emerging concepts in this disease.

**Abstract:**

Mixed-phenotype acute leukemias (MPAL) are rare in children and often lack consensus on optimal management. This review examines the current controversies and emerging paradigms in the management of pediatric MPAL. We examine risk stratification, outcomes of recent retrospective and prospective collaborative trials, and the role of transplantation and precision genomics, and outline emerging targets and concepts in this rare entity.

## 1. Introduction

Mixed phenotypic acute leukemias (MPAL) are rare hematological malignancies in children, accounting for less than 5% of pediatric acute leukemias [1,2,3,4]. MPAL are heterogeneous and can exhibit cross-lineage myeloid, B-lymphoid, or T-lymphoid antigen expression on a single blast population (biphenotypic) or have distinct single-lineage blast populations (bilineal) [3]. Due to phenotypic and genetic diversity, lack of well-defined diagnostic criteria, treatment resistance, and lineage switch, MPAL often present a diagnostic dilemma, and prove difficult to treat.

Biphenotypic MPAL are more common than bilineal MPAL. However, the true prevalence and survival can be difficult to determine given the various diagnostic criteria utilized, lack of a centralized review of cases, and treatment protocols that are based on results from retrospective studies. The Children’s Oncology Group Acute Leukemia of Ambiguous Lineage Task Force reported that routine institutional flow cytometry was insufficient for the diagnosis of MPAL in about 15% of children, which further highlights the diagnostic challenge faced by oncologists [5]. 

## 2. Classification of MPAL

Currently, MPAL are classified based on lineage-specific immunophenotypic markers determined by flow cytometry, immunohistochemistry, or cytochemistry and primary molecular alteration, and are considered by most co-operative groups to be high-risk leukemias [5]. The majority of MPAL present as B-lymphoid/myeloid (in about 2/3 cases), with a T-lymphoid/myeloid immunophenotype being the second most common presentation. Rarely, it can present as B-lymphoid/T-lymphoid or B-lymphoid/T-lymphoid/myeloid subtypes [6,7,8]. Myeloid-surface antigen co-expression does not appear to be prognostic [3]. The classification of MPAL also includes two distinct entities: MPAL with *KMT2A* (mixed-lineage leukemia or *MLL*) rearrangement and MPAL with t(9;22)(q34.1;q11.2); *BCR-ABL1*(Philadelphia chromosome positive or *Ph*+).

After MPAL were recognized as a distinct entity, numerous sets of diagnostic criteria were established, including the European Group for the Immunological Characterization of Leukemias (EGIL), and most recently the World Health Organization (WHO) 2016 system (updated from previous WHO 2008 classification), which is being increasingly utilized for MPAL diagnosis [6,9,10]. A hallmark of MPAL in the WHO classification scheme rests on the fact that other leukemia subtypes (i.e., AML-defining balanced translocations such as t(8;21) that frequently expresses multiple B-cell markers) must be excluded prior to the MPAL designation [11]. Given its more widespread adaption in clinical practice, the World Health organization (WHO) 2016 criteria are presented in Table 1 and Table 2. There are distinct differences between the classification schema, with the European Group for the Immunological Classification of Leukemias (EGIL) scheme generally considered to be more inclusive, which often leads to more acute leukemias being classified as MPAL, and a higher incidence of MPAL compared to the WHO classification. Weinberg et al. published a review looking at 7627 patients (both pediatrics and adults) with leukemia showing a mixed phenotype incidence of 2.8% using EGIL compared with 1.6% when using WHO 2008 criteria [3,11]. 

Further classification of MPAL can be implemented utilizing the mutational status of the leukemia. Most MPAL have an abnormal and often complex karyotype, with t(9;22) mostly identified in adult or older patients, whereas t(v; 11q23) (*AFF1* is the most common fusion partner of *MLL*) is primarily seen in infant B/myeloid MPAL [11]. Matutes et al. looked at 100 MPAL patients diagnosed using the WHO 2008 criteria and found that the most common abnormality was a complex karyotype in 32% of patients, t(9;22) in 20%, and normal karyotype in only 13% [8]. Less commonly, chromosome 1, 6, and 12 deletions; trisomy 4; and near-tetraploidy have been reported [5,6,8]. These mutations aid in subclassification but the therapeutic and prognostic importance is still under appreciated.

## 3. Molecular Studies in MPAL

The development of MPAL is thought to occur either through the progression of sequential mutations that drive a leukemia cell towards a multi-lineage phenotype or arise from primitive hematopoietic cells with multipotency that develop clonal evolution [12,13]. Evidence presented by Alexander et al. supports the primitive cell mechanism by displaying similar founder mutations and genomic imprinting throughout multiple subclones in MPAL patient samples [12]. 

A recent study out of St. Jude Children’s Research Hospital analyzed the largest cohort of over one hundred cases of pediatric MPAL which identified genomic patterns that showed several recurrent alterations common in ALL (i.e., *ETV6*), AML (i.e., *FLT3* and *RUNX1*), as well as those seen in both ALL/AML (i.e., *WT1* and *KMT2A*) [12]. Of note, there were unique mutational patterns noted between B/myeloid and T/myeloid MPAL. These mutations include alterations in the *JAK-STAT* pathway in T/myeloid MPAL (seen in 57% versus 23% of B/myeloid) and *ZNF384* rearrangements in B/myeloid, which leads to a higher *FLT3* expression than typical B-ALL or T-lymphoid/myeloid MPAL [12]. 

Recent publications have utilized comprehensive next-generation sequencing (NGS) to reveal the complex genetic landscape of MPAL which may further impact diagnosis, treatment strategies, and prognosis. The majority of B/Myeloid MPAL harbor *RAS* and epigenetic regulator mutations, with *NRAS* and *PTPN11* and *MLLT3*, *KMD6A*, *EP300*, and *CREBBP* being commonly altered [12]. Compared with T-ALL patients, T/myeloid MPAL demonstrate a lower frequency of alterations in *CDKN2A*/*B*, *NOTCH1*, and core transcription factors [12]. Most studies investigating the mutational landscape of T/Myeloid MPAL are limited by small numbers but have reported mutations in *ETV6*, *NOTCH1*, *WT-1*, and *FLT3,* as well as in epigenetic modifiers or regulators, *IKZF1* (Ikaros), and *JAK*/*STAT* signaling proteins [12,14,15,16,17]. Additionally, a retrospective trial out of Memorial Sloan Kettering showed that MPAL with a T-lymphoid component had higher expression of mutations in *PHF6* (seen in younger patients) and *DNMT3A* (seen in older patients), which is line with previous work [7,18]. 

Additional studies with a limited number of patients have been conducted in adult patients, further elucidating the diverse genetic landscape of MPAL. A study conducted by Takahashi et al. looked at thirty-one adult MPAL patients and saw unique mutational patterns between MPAL subtypes (particularly B-lymphoid/myeloid and T-lymphoid/myeloid) [19]. Of note, the commonly seen rearrangement in *ZNF384* seen in the St. Jude pediatric population of B-myeloid MPAL was not seen in the adult cohort and would limit *FLT3* as a target of therapy [7,19]. 

In summary, the studies outlined above elucidate the heterogenous biology and lineage plasticity of MPAL, and future work may help identify unique targetable molecular lesions, using precision genomics, in patients with poor response to therapy. 

## 4. Lineage Switch in MPAL

Lineage switch is a phenomenon in which a lymphoid leukemia converts to a myeloid leukemia (more common), but the opposite switch has also been reported [20]. This phenomenon is of particular interest in MPAL, is more commonly reported in *KM2TA* (*MLL*)-rearranged leukemias, and often complicates management of these patients [21]. 

A review article by Hu et al. has elegantly outlined that the mechanisms of lineage switch are not fully understood and presents several complimentary hypotheses [20]. One hypothesis is that leukemic clones involved in lineage switches and ambiguous leukemias may in fact be derived from multipotent hematopoietic cells [20,22]. This concept is further supported by methylation studies performed by Alexander et al. [12]. Another potential mechanism of lineage switch involves leukemic cell reprogramming by transcription factors. It is well known that transcription factors are important for normal hematopoiesis and loss of transcriptional regulation can lead to leukemia development [20,23]. Overexpression of *Pu.1* (an important transcription factor for myeloid regulation) has been shown to cause lineage switch in mouse models, as well as in a case of myeloid switch adult leukemia [20,24,25]. *Pax 5* is considered an important transcription factor for B-lymphoid lineage differentiation with high levels seen in B-ALL and lower levels seen in MPAL following a lineage switch (i.e., after CD19-directed CAR-T cell therapy) [20,26]. The pressure from the cytokine microenvironment may also play a pivotal role in lineage ambiguity/switch. B-ALL cells have been reprogrammed into myeloid cells (non-leukemic macrophages) through exposure to pro-myeloid cytokines including IL-3 and GM-CSF in in vitro models, which suggests microenvironmental pressure may impact leukemia lineage [27]. The importance of transcription factor and tumor microenvironmental changes is not specific to MPAL but the importance has also been seen in mature B-cell lymphomas (including the importance of IL-6 and *PAX5*) [28,29]. A simplified diagram highlighting the impact of transcription factors and cytokines is shown in Figure 1.

Finally, treatment-related clonal selection pressure may lead to lineage switch or development of a mixed-lineage leukemia. This is of particular interest with the increased use of B-ALL-directed immunotherapy, which has been linked to lineage switch, particularly in *KMT2A*-rearranged leukemias [20,21,26,30,31]. It is unclear if selective killing of a CD19-positive clone with subsequent expansion of a more myeloid clone is to blame or if transcriptional alteration (upregulation of *Pu.1* and downregulation of *Pax5*) could play a role [26]. A simplified schema outlining potential mechanisms of lineage switch in acute MPAL is presented in Figure 2. 

## 5. Treatment of MPAL

Historically, MPAL have inferior outcomes and a high risk of induction failure, compared with ALL/AML, and are treated per high-risk leukemias protocols [32]. Poor prognostic factors include: an older age at diagnosis, higher white blood cell (WBC) count at presentation, T-lymphoid/myeloid phenotype, adverse cytogenetics (such as a *KMT2A*/*AFF1* rearrangement), extramedullary disease at diagnosis, and MRD positivity [11,33,34]. There have been various chemotherapy approaches for the treatment of MPAL including acute ALL, AML, and hybrid ALL/AML (such as FLAG (fludrabine, cytarabine, granulocyte-stimulating factor)-IDA (idarubicin) with vincristine and prednisone (VCR-PRED) or hyper-CVAD (cyclophosphamide, vincristine, doxorubicin, and dexamethasone) regimens) [8,35,36]. Optimal therapy remains a subject of controversy and differences between adult and pediatric treatment approaches are often striking [8,37]. 

St. Jude Children’s Research Hospital reviewed outcomes of 35 pediatric MPAL patients (treated from 1985 to 2006) and reported that the majority (65%) of the patients received AML induction chemotherapy with cytarabine, daunomycin, and etoposide, while 35% of the patients received ALL four-drug induction [15]. In this review, of the patients treated with upfront AML therapy, 12/23 (52%) achieved complete remission compared with 10/12 (83%) with ALL therapy. Interestingly, 8/10 (80%) of patients who did not achieve complete remission (CR) with AML therapy went into remission after being switched to ALL-like induction therapy [15]. Long-term survival was achieved in 17 out of 35 patients (5 patients treated with AML therapy and 12 patients treated with ALL therapy either upfront or after initial AML failure). The same study compared historical survival rates, comparing MPAL with standard ALL therapy to AML therapy. The 5-year survival estimates for MPAL (combined B/myeloid and T/myeloid) were 47.8% ± 11.5%, similar to that of AML (56.5% ± 3.5%), but were significantly less than patients receiving ALL therapy at 84.6% ± 1.1% [15]. 

Previous studies on adults have shown that the historical 4-year survival for adult MPAL is less than 10% [8]. A case series of 100 MPAL patients by Matutes et al. further showed that older patients had inferior survival compared with patients less than 15 years of age [8,38]. In this case, series median survival in adults was 11.8 months compared with 139 months in children. A similar pattern was seen in patients treated with ALL therapy, with a median survival of 139 months compared with 11 months with AML therapy and 3 months with a hybrid approach [8].

However, more recently, due to improved diagnostic criteria and genomic techniques, and the observation that ALL-like regimens in both pediatric and adult populations are associated with superior treatment response, the treatment landscape has clearly changed [5,35,39,40,41].

Several other studies have also investigated optimal upfront therapy for patients with MPAL, and support that notion that ALL treatment regimens tend to lead to better overall survival than AML-based regimens [1,7,11,16,38,39,42,43,44]. The BFM group international pediatric cooperative study (AMBL2012) demonstrated a superior outcome when patients were treated with ALL primary therapy with a 5-year event-free survival (EFS) of 80% ± 4% compared with AML therapy (36% ± 7.2%) or hybrid ALL/AML therapy (50% ± 12%) [36]. In particular, ALL/AML hybrid approach in *KMT2A*-rearranged patients resulted in a subpar 5-year EFS of only 28% [36]. A 2018 systematic review from Maruffi et al. looked at over 1300 adults and children diagnosed with MPAL and showed that ALL induction regimen was more likely to lead to remission (OR = 0.33) and improved overall survival (OR = 0.45) compared to AML-like treatment protocols, or an even worse outcome associated with hybrid regimens [1]. Additionally, a recent multi-center analysis showed that ALL induction therapy was able to achieve MRD negative remission by the end of induction in a majority of patients (70%) [40]. Even in studies showing similar survival benefits between ALL and AML/hybrid regimens, ALL regimens tend to lead to overall decreased morbidity, due to less toxicity, compared with AML or hybrid-based treatments [35]. Lumbar puncture is recommended at diagnosis to determine central nervous system (CNS) status for all MPAL patients, as frequency of CNS involvement is higher [7,45]. CNS-directed intrathecal chemotherapy is administered, similar to treatment protocols for acute leukemia with CNS involvement. 

A recent review from Children’s Healthcare of Atlanta, highlights that patients with B/myeloid MPAL with isolated MPO expression might in fact be a unique entity, and have a more favorable response to ALL therapy [46]. In this review, patients with B/myeloid with isolated MPO had an overall survival rate at 3 years of 100% compared with 63.1% with other MPAL, and interestingly, the degree of MPO positivity was not prognostically relevant [46]. This highlights that the detection of a B/MPAL phenotype with isolated MPO expression is important, and may allow for better prognostic information and treatment decisions.

MPAL with *KMT2A* (*MLL*) rearrangement, are more common in children than adults, are typically bilineal (lymphoblasts and monoblasts) but rarely biphenotypic, and are prone to lineage switch [21,24,30,37]. The fusion partner of MLL1 is a determinant of the leukemic phenotype [47]. For example, *MLL-AF4* is predominantly associated with a lymphoid phenotype and *MLL-AF9* with a myeloid phenotype [47]; however, the role of fusion partners in determining MPAL phenotype or lineage switch remains unclear [48]. Compared with *Ph*+ MPAL, the *MLL*+ MPAL patients have significantly inferior survival odds (HR = 10.2, *p* < 0.001) [49], and are transplanted, upfront, if induction failure occurs, or at relapse [11,50,51]. New emerging targets blocking the *MLL* fusion complex are under evaluation, include *menin* [52], disruptor of telomeric silencing 1-like (*DOTL1*) inhibitors [53], and spleen tyrosine kinase (*SYK*) inhibitors [54].

## 6. Treatment of *Ph*+ MPAL

*Ph*+ acute ALL, with t(9;22)(q34:q11.2), constitute 25% of MPAL, and are more common in adults than children, and often present with a B/myeloid phenotype and a p190 *BCR-ABL* transcript and have a high incidence of CNS involvement [3,8,11]. With the advent of successful TKI therapy directed against *BCR-ABL*, the treatment paradigm has shifted, and outcomes have improved significantly [49]. An ALL-chemotherapy backbone combined with a TKI is increasingly utilized, with comparable outcomes to Ph+ non-MPAL ALL [49]. However, *Ph*+ MPAL must be distinguished from CML with mixed blast crisis, as the latter is treated with AML-like therapy followed by allogeneic transplant. CML is usually associated with a *p210* isoform, and MPAL usually express the *p190*, although each could be associated with either isoform. Prior history of a chronic phase or accelerated phase may also distinguish these entities [55,56]. 

## 7. Role of Hematopoietic Stem Cell Transplantation

The role of hematopoietic stem cell transplantation (HSCT) in MPAL remains controversial and recommendations may vary based on the age of the patient being considered. Several adult retrospective trials show that survival is poor with chemotherapy alone and that there is possible benefit for the use of HSCT if a patient is able to obtain complete remission (CR) [1,50,51,57,58]. Munker et al. analyzed a large adult cohort for HSCT outcomes in MPAL: the 3-year overall survival was 56%, which is an improvement over the historical SEER data for patients older than 20 years of age, which tend to be 20–40% [59]. On the other hand, several studies have shown there is no clear benefit for the use of HSCT in pediatric MPAL, and that most pediatric MPAL are able to be treated with ALL-based chemotherapy regimens alone [17,58,59]. In pediatrics, HSCT is typically only pursued in patients with induction failure, a high minimal residual disease (MRD) >5% at end of induction (EOI) or persistent MRD positivity at the end of consolidation (EOC), or after lineage switch [5]. There is a survival advantage if total body irradiation is incorporated into the transplant conditioning (similar to ALL), when compared with other preparative regimens [57].

Relapsed MPAL post transplantation remains a major challenge, and generally there is no consensus for further management of these cases. Patients treated with an ALL approach appear to be less likely to relapse than patients treated with an AML therapy approach. In addition, the remission durations of relapsed patients tend to be shorter, while age, lineage switch, and higher white counts are not predictive of worse outcomes per se [60].

## 8. Assessing Response/Impact of MRD

MRD by flow cytometry or next-generation sequencing (NGS) is a well-established technique to assess therapy response and prognosis in acute leukemias [32,61]. MRD positive disease following intense chemotherapy has been shown to be a powerful prognostic indicator in pediatric and adult B-ALL [62]. MRD evaluation in MPAL presents unique challenges, due to variable antigen expression, potential for lineage switch, the lack of validated MRD assays assessing the dual expression or bilineal blasts, and the lack of established thresholds to assess for adequate therapy response [7]. 

Despite these challenges, MRD positivity has been linked to worse outcomes in MPAL. Patients with EOI MRD ≥ 5% performed poorly in the iBFM AMBL2012 trial with a 5-year EFS of less than 50% [36]. Furthermore, this trial demonstrated that patients who achieved an MRD < 0.01% by the EOC had excellent outcomes with an overall survival of near 90% [5,36]. An additional multi-center review of almost 100 pediatric MPAL patients showed that MRD positivity at EOI was associated with worse EFS and OS (HR = 6.00 and HR = 9.57 respectively) [40]. This flow-cytometry-based MRD-directed approach has been incorporated by the Children’s Oncology Group (COG) to identify MPAL treatment failures [5]. NGS-based MRD assays or identification of mutational hotspots may assist in tracking and assessing one or more populations of MPAL blasts and requires further study [5]. A high end of induction or consolidation MRD may necessitate augmentation of therapy, a switch from ALL to AML directed therapy, and a consolidative hematopoietic stem cell transplant [1]. 

## 9. Novel Approaches 

With the growing molecular/genetic understanding of MPAL, this opens the door for potential targeted, novel therapeutic approaches. These targeted therapies include those already being successfully utilized in the treatment of B-ALL, including tyrosine kinase inhibitors for *Ph*+/*Ph*-like disease [63,64,65] and *FLT3* inhibitors used in *MLL*-rearranged disease [12,66]. *BCR-ABL* translocations had previously been shown to have poor outcomes in MPAL, but with the addition of TKI therapy to an ALL backbone they have outcomes similar to *Ph*+ B-ALL [65]. *FLT3* inhibition is an attractive option, particularly given the data from St. Jude showing the increased prevalence of *ZNF384* rearrangement leading to upregulated *FLT3* expression in B-lymphoid/myeloid MPAL. Pre-clinical and clinical research has been conducted in *KMT2A*-rearranged leukemia (particularly in infants) and has shown increased sensitivity to both FLT3 and RAS inhibition, which could be translated into MPAL patients harboring such mutations [67,68,69]. Novel agents such as *BCL-2* inhibitor venetoclax, in combination with hypomethylating agents, have been used to induce remission in MPAL [70,71,72]. Given the unique biology of MPAL, engineering hematopoietic stem cells harboring altered transcription factor profiles could mimic a pre-clinical model and allow the expanded testing of novel target agents, using a precision-medicine approach.

An expanding area of research includes the use of immunotherapy for the treatment of hematologic malignancies. Immunotherapies, including bispecific T-cell engagers (BITE) and chimeric antigen receptor T-cells (CART), have been successfully utilized for the treatment of CD19+ relapsed/refractory B-ALL [73,74,75,76]. There is growing evidence that these therapies can also be utilized in the treatment of MPAL. Recent case reports and small retrospective reviews support the use of CD19 BITE cells and CART cells as efficacious treatments for refractory MPAL [77,78,79,80]. A recent case series from Bartram et al. out of the United Kingdom presented three pediatric cases of refractory MPAL that successfully utilized Blinatumomab as a bridge to transplant, including in a patient with a significant CD19-negative blast component [81]. Given the presence of different lineage antigens in MPAL, it is thought that bi-specific immunotherapy options might be more efficacious. Pre-clinical data have shown that dual-targeting triplebody therapy directed against CD33/CD3/CD19 leads to selective and effective lysis of biphenotypic B/myeloid cells expressing CD19 and CD33 [82,83]. A recent case report successfully utilized two immunotherapy options targeting separate antigens (Blinatumomab for CD19 and Gemtuzumab for CD33) to successfully treat a refractory, KMT2A-rearranged infant MPAL [37]. Table 3 summarizes clinical reports that highlight the successful use of immunotherapy in MPAL. Despite potential success, the use of direct/immunotherapy-based treatment strategies for MPAL proposes unique challenges, including selective pressure towards treatment resistance and lineage switch, requiring further studies.

## 10. Conclusions and Future Directions

MPAL are a heterogeneous group of leukemias that often have a complex phenotype/genetic basis and historically have been difficult to diagnose and treat. Despite recent advances in the diagnosis criteria and treatment landscape of MPAL, there is still much to learn about this unique subset of acute leukemias. As most current treatment recommendations are based on retrospective studies, prospective clinical trials standardizing the treatment regimens and utilizing MRD for assessing treatment response, such as the ongoing COG trial AALL1732, are urgently needed to solidify a uniform approach for the management of MPAL.

## Figures and Tables

**Figure 1 cancers-13-04658-f001:**
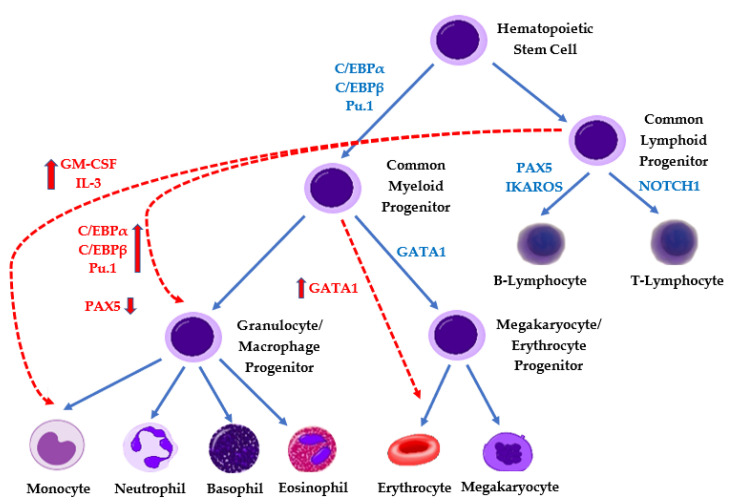
Regulation of lineage commitment in hematopoiesis. Solid lines/blue text indicate regulators of normal hematopoiesis, and red lines (---) and text highlight aberrant regulation implicated in MPAL pathogenesis.

**Figure 2 cancers-13-04658-f002:**
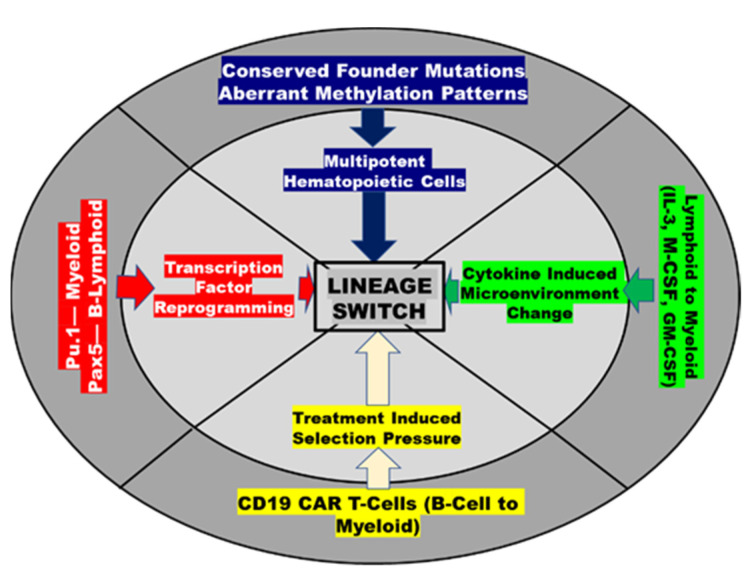
Potential mechanisms causing lineage switch in MPAL.

**Table 1 cancers-13-04658-t001:** Criteria for lineage assignment in mixed phenotypic acute leukemia.

**Lineage Assignment Criteria**
Myeloid Lineage
MPO+ (Flow cytometry, immunohistochemistry, or cytochemistry) or Monocytic differentiation (at least two of the following: nonspecific esterase cytochemistry, CD11c, CD14, CD64, lysozyme)
**T-Lymphoid Lineage**
Strong * cytoplasmic CD3 (with antibodies to CD3 ε chain) or Surface CD3
B-Lymphoid Lineage
Strong * CD19 with at least 1 of the following strongly expressed: CD79a, cytoplasmic CD22, or CD10 or Weak CD19 with at least 2 of the following strongly expressed: CD79a, cytoplasmic CD22, or CD10

* Strong is defined by equal or brighter expression than normal B or T cells in the sample.

**Table 2 cancers-13-04658-t002:** WHO 2016 criteria for acute leukemia of ambiguous lineage.

Acute Undifferentiated Leukemia
Mixed-phenotype acute leukemia (MPAL) with t(9;22)(q34.1;q11.2); BCR-ABL1
MPAL with t(v;11q23.3); KMT2A rearranged
MPAL, B/myeloid, NOS
MPAL, T/myeloid, NOS

**Table 3 cancers-13-04658-t003:** Clinical reports utilizing immunotherapy to treat MPAL (FLA: fludrabine, cytarabine; IDA: idarubicin).

Reference	MPAL Patients	Treatments	Outcomes
Bartram et al. [81]	Patient 1: 6-month-old female with B/Myeloid MPAL with KMT2A deletion Patient 2: 10-month-old male with B/Myeloid, KMT2A-USP2, and FLT3 ITD Patient 3: 72-month-old female with B/Myeloid MPAL and KMT2A-ARHGEF12	ALL induction followed by FLA-IDA (myeloid regimen) with CD19+ MRD, and Blinatumomab followed by HSCT >10% disease after ALL induction (CD19+, CD19-, populations): Blinatumomab and FLA-IDA prior to HSCT ALL induction followed by Blinatumomab and high-dose Ara-C prior to HSCT	CR for 24 months post HSCT CR for 8 months post HSCT CR 5 months post HSCT
Li et al. [77]	29-year-old male with B/myeloid CD19+ MPAL and SET-NUP14 fusion gene transcript	ALL induction followed by hybrid Consolidation (hyper-CVAD-B and hyper-CVAD-A) and HSCT For Relapse#1: received CD19 Directed CART therapy x2 (donor-derived)	Relapse#1 after 6 months Relapse#2: 24 months following first infusion of CAR T-Cells despite persistent CD19 CAR T-Cells at 8 months
Durer et al. [77]	51-year-old female with B/myeloid CD19+ MPAL	ALL induction followed by HSCT but relapsed after 3 months. Received Blinatumomab and donor lymphocyte infusions (x 4 cycles)	Attained CR after one cycle and maintained CR for 15 months
El Chaer et al. [78]	Patient 1: 39-year-old male with Ph+ B/myeloid MPAL Patient 2: 55-year-old female with B/myeloid	Induction with Cytarabine, Daunorubicin, and Dasatinib followed by Blinatumomab with HSCT HyperCVAD (held due to toxicity), and later received 3 cycles of Blinatumomab followed by HSCT	CR at 6 months CR at 14 months
Brethon et al. [37]	4-month-old female with B/myeloid MPAL with KMT2A-AFF1	Interfant-06 protocol with refractory disease; received Blinatumomab and Gemtuzumab followed by HSCT, followed by CART for relapse	Relapsed 11 months post HSCT; then achieved CR for 12 months post CART

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
