# Peer review of "Pediatric Mixed-Phenotype Acute Leukemia: What’s New?"

_cancers, 2021, doi:10.3390/cancers13184658_

Round 1

Reviewer 1 Report

This manuscript, written by Dr. Sandeep Batra, with the title of “Pediatric mixed Phenotype Acute Leukemia: What’s new?” is a review that focuses on mixed phenotype acute leukemia (MPAL). MPAL refers to acute leukemia that displays an ambiguous pattern of antigen expression (ie, reflecting more than one hematopoietic lineage), to a degree that it cannot be unequivocally assigned to one lineage. The manuscript is well written, it is easy to read, has 1 table, 1 figure, and has enough references. Before publishing this review, the authors could address the following minor comments:

  1. Page 1 of 10, lines 41-42. Could you please format the name of gene in italics? Please also use italics for the rest of the manuscript.
  2. Page 1 of 11, line 42-43. Maybe you could write the entity as “Mixed phenotype acute leukemia (MPAL) with t(9;22)(q34.1;q11.2);BCR-ABL1”.
  3. Page 2 of 10, lines 70-71. Could you please add in the text of Table 1
  4. Page 2 of 10, line 71. Here, after Table 1, I would add another table. To be specific, the table of the World Health Organization (WHO) classification of myeloid neoplasms and acute leukemia. Or at least a table with the WHO classification of Acute leukemias of ambiguous lineage, as follows: (1) Acute undifferentiated leukemia, (2) Mixed phenotype acute leukemia (MPAL) with t(9;22)(q34.1;q11.2);BCR-ABL1, (3) MPAL with t(v;11q23.3); KMT2A rearranged, (4) MPAL, B/myeloid, NOS, and (5) MPAL, T/myeloid, NOS.
  5. Page 2-3 of 10, lines 72-109, section 3 “molecular studies in MPAL”. Could you please summarize the information in a table? (if this is possible). Were the criteria for “high confidence” mutations similar between the different publications? Is there a postulated disease pathogenesis interpretation of these recurrent mutations (e.g. which pathways were affected?)
  6. Pages 3-4, lines 110-138, section 4 “Lineage Switch in MPAL”. Maybe the authors could show a diagram with the different stages of the hematopoiesis, and its relationship with MPAL.
  7. Pages 3-4, lines 127-130. The authors wrote “The pressure from the cytokine microenvironment may also play a pivotal role in lineage ambiguity/switch. B-ALL cells have been reprogrammed into myeloid cells through exposure to pro-myeloid cytokines including IL-3 and GM-CSF.”. Do the authors suggest that B lymphocytes can be reprogrammed into myeloid cells because of the immune microenvironment? Does this change happen in other types of hematological neoplasia, including mature B cell lymphomas?
  8. Page 4 of 10, lines 142-144, “Historically, MPALs that lack the Philadelphia chromosome (Ph) negative (Ph-) have 142 inferior outcomes, and a high risk of induction failure, compared to ALL/AML and are 143 treated per high risk leukemias protocols.”. Could you please comment on the prognostic significance of Philadelphia chromosome-positive (Ph+) as a poor prognostic feature in the era of routine use of tyrosine kinase inhibitors?
  9. Could you please comment on the role of lumbar puncture at diagnosis?
  10. The authors could add histological images, or flow cytometry figures.
  11. This review has only one table. The authors could make more tables summarizing the most relevant information of the different sections.
  12. Before the conclusion and future directions, the authors could write a summary and recommendations of treatment, possibly in the form of an algorithm.
  13. The authors focused on pediatric MPAL. What about adult MPAL?

Reviewer 2 Report

The topic of pediatric MPAL has a great clinical interest and includes many important clinical research works. The authors commented a lot of retrospective studies that have been determinant to improve risk stratification and lineage assignment, according to morphology, surface markers and genetics features.

The review is well written and I propose the publication after minor revisions:

 1- In the section “treatment of MPAL Ph- “. I propose to add some recent target drugs and some important references in this topic. It is known that the presence of MLL is correlated with FLT3 high expression and the leukemic cells from infants with MLL were significantly highly sensitive to the FLT3 inhibitors (PMID: 15956279). In addition, it has been demonstrated that some MLL fusion genes functionally correlated with RAS pathway activation as well as with the presence of RAS mutation (PMID: 29056538). As consequence MLLr displayed high response to drugs targeting RAS pathway (PMID: 24695851). The reference about DOT1L inhibitor is referred to a research work performed in the adult cohort. 

2- I propose to add section exploring the knowledge about biology and leukemogenesis of MPAL. Indeed, there are many evidences supporting the hypothesis that specific genomic alteration occurs at an early stage of hematopoietic development and it is determinant to maintain multi-lineage phenotypic potential. Indeed, it has been demonstrated that distinct phenotypic subpopulations harbored identical DNA mutations (PMID: 30209392). As consequence, in the era of precision medicine, they need a valid preclinical model to test target drugs. Engineered induced pluripotent stem cells can recapitulate the early stage of hematopoietic development and leukemogenesis process and can be a new perspective to add.
